# Prevalence of Anxiety and Burnout, and Coping Mechanisms among Clinical Year Medical Undergraduate Students in Universiti Kebangsaan Malaysia Amidst the COVID-19 Pandemic

**DOI:** 10.3390/ijerph192013010

**Published:** 2022-10-11

**Authors:** Ke Ran Tee, Abdul Samat Ismail, Yong Heng Ang, Hidayah Husna Hishamuddin, Vinneeshah Jacob Paul, Azimatun Noor Aizuddin, Ida Zarina Zaini

**Affiliations:** 1Department of Emergency Medicine, Faculty of Medicine, Universiti Kebangsaan Malaysia Medical Centre, Jalan Yaacob Latif, Bandar Tun Razak, Kuala Lumpur 56000, Malaysia; 2Department of Community Health, Faculty of Medicine, Universiti Kebangsaan Malaysia Medical Centre, Jalan Yaacob Latif, Bandar Tun Razak, Kuala Lumpur 56000, Malaysia

**Keywords:** coronavirus disease 2019, anxiety, burnout, coping mechanism, undergraduate medical students

## Abstract

This study aimed to determine the prevalence of anxiety and burnout, and the coping mechanisms among clinical year undergraduate medical students in Universiti Kebangsaan Malaysia (UKM) during the coronavirus disease 2019 (COVID-19) pandemic. In total, 378 clinical year undergraduate medical students in UKM participated in this cross-sectional study from May to July 2021. A self-administered questionnaire consisting of questions on the participant’s sociodemographic data and items from the DASS-21, CBI, and Brief-COPE was distributed. Chi-square and Spearman’s correlation tests were used to calculate the correlation coefficient between both anxiety and burnout, and coping mechanisms. The prevalence of anxiety and burnout were 44.2% and 22.2%, respectively. There was a significant difference in the percentage of students with extremely severe anxiety in the presence and absence of burnout, 23.8% vs. 4.8% (*p* < 0.001). Among the three coping mechanisms, avoidant coping had a significant positive moderate correlation with both the presence of anxiety (*r* = 0.3966, *p* < 0.001) and the presence of burnout (*r* = 0.341, *p* < 0.001). Meanwhile, coping that was neither approach nor avoidant had a positive weak correlation with the presence of burnout (*r* = 0.176, *p* = 0.001). The prevalence of anxiety and burnout was concerning. Increased anxiety and burnout among students may negatively impact aspects of their personal, professional, and academic lives. Early recognition and preventive measures should be emphasised to prevent negative ramifications.

## 1. Introduction

Anxiety is a psychological and physiological condition, characterized by cognitive, somatic, emotional, and behavioural components. It acts as a protective factor against threatening situations, while burnout entails feelings or experiences of exhaustion, cynicism, and lack of professional efficacy [1,2]. Anxiety is normal when it is intermittent and expected, based on certain stressful events or situations. However, prolonged anxiety might result in psychological distress affecting an individual’s everyday functioning.

Research focusing on the concept of burnout became more prevalent during the 1970s where researchers investigated a phenomenon resulting from excessive work demands, later known as “burnout”. The sources of burnout can include workload, control, reward, community, fairness, and values. Burnout can result from a mismatch between the individual and the work environment. Burnout is classified into three types, namely which is personal burnout, work-related burnout, and client-related burnout. Personal burnout is defined as the degree of physical and psychological fatigue, and exhaustion experienced by a person regardless of their working status. Work-related burnout measures the same features, which include the physical and psychological fatigue, and exhaustion experienced by the person but perceived as strictly related to his/her work. Client-related burnout measures the degree of physical and psychological fatigue, and exhaustion the person undergoes at work that is related to his/her clients. A “client” is a broad concept covering patients, children, students, and residents [1,2].

The theoretical framework for this study was based on the cognitive theory of coping developed by Folkman and Lazarus. This theory was selected because it links the constructs central to this study. Coping was defined as a change in cognitive and behavioural efforts to manage specific external and/or internal demands that are seen as exceeding the resources of the person. This important theory says that an individual’s knowledge acquisition can be directly related to observing others’ examples in social interactions, media influences, and experiences. Coping is a survival technique because it is a result of replicating others’ actions. There are three meta-theoretical assumptions: transaction, process, and context. Lazarus and Folkman’s transactional model describes the interaction between the individual and the stressful event, which is particularly obvious in the individual’s assessment of the problem being faced. According to this interactive model, the individual goes through two different processes which are important to the outcome of the problem. The first is the cognitive assessment, which refers to the way the situation relates to the individual. The second refers to how to deal with the problem [3].

In our study, we evaluated the coping mechanisms used by clinical year medical undergraduate students in the presence or absence of anxiety and burnout during the COVID-19 pandemic by using the Brief-COPE (Coping Orientation to Problems Experienced) tool. For our study, coping refers to the process by which a student successfully manages stressful events that are seen as exceeding the available resources. The Brief-COPE tool is the abbreviated version of the COPE Inventory and assesses dispositional as well as situational coping efforts. The inventory is the tool most widely used to measure coping. It is a shortened version of the full 60-item (16 scales) COPE inventory developed by Carver et al. [4]. The 28-item Brief-COPE (consisting of 14 subscales) has acceptable psychometric properties and has been used extensively to examine the relationship between various coping strategies and psychological outcomes in various populations.

The measurement models of Brief-COPE can group the 14 subscales into two-category or three-category models [5]. In our research, we used the three-category version of the Brief-COPE measurement model. The categories are approach coping (e.g., acceptance, positive reframing, active coping, planning, emotional support, and informational support), avoidant coping (e.g., self-distraction, positive reframing, venting, self-blame, behavioural disengagement, denial, and substance abuse), and neither approach nor avoidant coping (e.g., religion and humour) [6]. Each individual responds to stress differently, depending on their personality traits and characteristics. An individual engages in these coping strategies to manage the distress resulting from a new stressor. Approach coping strategies lead an individual toward a stressor, with the intention of resolution. Conversely, avoidant strategies lead one away from a stressor, with the intention of reducing the negative impact on the self. Approach strategies have long been seen as the more adaptive of the two, and avoidant strategies are seen as almost entirely nonadaptive [7,8].

The UKM Doctor of Medicine (MD) undergraduate course aims to produce well-rounded medical professionals who can serve society in varied work environments and who are well-positioned for further postgraduate training. Before the pandemic began, the UKM undergraduate medical course’s structure included ward rounds, tutorials, lectures, and seminars to nurture skills in problem solving, methods of inquiry, critical thinking, and independent learning strategies [9,10].

During the COVID-19 pandemic, the teaching and learning (T&L) methods were adapted to the current circumstances. Digital learning and clinical experience simulation software were introduced to help the students continue their studies despite restricted entry to clinics, wards, and academic buildings. Digital learning is a process of integrating technology-mediated synchronous and asynchronous approaches including assessments, assignments, and tutoring, and it enables learning without any time and location restrictions. The migration of face-to-face learning to synchronous or asynchronous digital learning with a period of transition and adaptation, was associated with challenges such as poor connectivity and a lack of the knowledge, tools, and infrastructure [11].

The medical students faced difficulties in adapting to the “new normal”, e.g., the nationwide quarantine measures, the faculty’s digital learning initiative, and reduced clinical exposure [12]. The pandemic and its consequent measures have also increased the prevalence of psychological distress in other parts of the world, such as in Iran [13], Spain [14], and China [15,16,17,18]. The risk of developing anxiety was also seen to be higher among medical students compared with the general population, especially in Asia. This could be attributed to the high-pressure environment where they train, as well as their underlying personality traits. An example of such traits is maladaptive perfectionism, which may cause higher levels of stress when there is a disruption of routine, as in the case of the COVID-19 pandemic and its associated public health measures [19,20]. A study carried out in Cyprus found that the mental health of medical students was seen to deteriorate with the migration to digital learning in view of the COVID-19 pandemic [21]. Other stressors linked to the pandemic included threat of the disease itself, the increased worry regarding personal health status and that of family members, the lack of timely and transparent information disclosed by the authorities, the loss of personal freedom, and social distancing [22].

With this background, this study aimed to determine the prevalence of anxiety and burnout, and the associations between anxiety and burnout, between anxiety and the coping mechanisms implemented, and between burnout and the coping mechanisms implemented in UKM’s clinical year undergraduate medical students amidst the COVID-19 pandemic. We hypothesized that the anxiety level among clinical year medical undergraduate students would be higher compared with pre-clinical year medical undergraduate students. Therefore, the clinical year medical undergraduate students were selected for this study. We hoped to obtain baseline information regarding anxiety, burnout, and coping mechanisms among the clinical year undergraduate medical student through our study. The result from this study might be helpful for future research or randomized controlled trials with better interventions.

## 2. Methodology

### 2.1. Participants

A cross-sectional study was conducted among clinical year undergraduate medical students in the Faculty of Medicine, UKM. The study was carried out between May and July 2021 during the 2020/2021 academic session. This population group was selected because limited data have been collected on clinical year undergraduate medical students who are studying in Malaysia. All clinical year undergraduate medical students (Year 3 to Year 5) who gave consent were included in this study. The exclusion criteria were incomplete questionnaires and those diagnosed to have an anxiety disorder(s) prior to the COVID-19 pandemic. Self-administered questionnaires and informed consent forms were distributed during face-to-face sessions.

A universal sampling method was used In this study, and the sample size was calculated using Kish’s [23] formula and the reference values from a meta-analysis by Quek et al., which reported that the prevalence of anxiety and burnout among medical students (P) is around 33.8% [20]. To achieve the sample size, the absolute precision (D) of 5% of the true proportion at 95% confidence was used. The Z-value was a constant of 1.96. The required sample size (*n*_1_) was estimated to be 344.

With the expected response rate set at 90%, this was further inflated by 10% to accommodate any missing data. After calculation, a sample size (*n*) of 378 was needed to adequately estimate the population’s prevalence with good precision.

### 2.2. Materials

The questionnaire package consisted of two parts, Part A and Part B. Part A consisted of the participant’s information sheet and a mental health pamphlet. The mental health pamphlet consisted of information on anxiety and burnout, and the contact details of mental health helplines. The purpose of this pamphlet was to provide additional information and resources for obtaining mental health support in the hope of helping any participants who may be afflicted.

Part B consisted of the informed consent form, sociodemographic data, and the anxiety subscales from the Depression, Anxiety, Stress Scale-21 (DASS-21), the Copenhagen Burnout Inventory (CBI), and the Brief Coping Orientation to Problems Experienced (Brief-COPE) inventory. Anxiety can be categorised into be mild, moderate, or severe by using the DASS-21. There were 21 questions in total, which consisted of components for the three domains, namely depression, anxiety, and stress. The items have responses in terms of frequency along a 4-point Likert scale. The responses were then added to produce a total score for the scale [24]. The CBI has three subscales: personal, work-related, and client related. The items have responses in terms of frequency along a 5-point Likert scale [25]. The CBI can accurately conceptualize burnout as a fatigue phenomenon with good reliability and validity, and can distinguish between work and personal factors. It has also been found to be suitable for use with health professionals because of the inclusion of client-related burnout [26]. The total score of each subscale was calculated and the mean was taken to represent severity of burnout in each subscale. Brief-COPE is used widely to assess coping mechanisms [6,27]. It consists of 28 items with 14 scales, and the items have responses in terms of frequency along a 4-point Likert scale, and the mean and standard deviation for the 2 related items are calculated to represent the scale.

The English and Malay versions of DASS-21 have good Cronbach’s alpha coefficients for anxiety, namely *α* = 0.74 and 0.85, respectively [24,28]. The English and Malay versions of the CBI subscales have Cronbach’s alpha coefficients of *α* = 0.85–0.87 and 0.83–0.87, respectively [25,29]. Brief-COPE has a Cronbach’s alpha coefficient of 0.88 for active coping and 0.81 for avoidant coping for the English version, and a total Cronbach’s alpha coefficient of 0.83 for the Malay version [27,30].

### 2.3. Data Analysis

All the data collected were cleaned and analysed using Statistical Package for Social Sciences (SPSS) Version 26. A descriptive analysis was performed to describe the sociodemographic characteristics, psychological distress, and coping mechanisms implemented by the respondents. Further analysis using the Chi-square test was used to examine the association between anxiety and burnout. Meanwhile, Spearman’s correlation test was conducted to delineate the association between coping strategies and psychological distress. *p*-values less than 0.05 were considered to be significant for all analyses in this study.

### 2.4. Ethical Approval, Considerations, and Declaration

This study was carried out in accordance to the ethics of clinical research, where the autonomy, integrity, and safety of the study subjects were guaranteed. Information sheets consisting of an explanation of the study’s objectives and the informed consent form were obtained from all respondents prior to completing the questionnaire. All data retrieved were kept confidential and anonymous. Due to this, the researchers chose not to communicate with participants who were at risk of or who were experiencing anxiety and/or burnout. Instead, the researchers chose to convey information on the recognition of anxiety and burnout, and a non-exhaustive list of resources to seek help from to the participants. This was achieved via a pamphlet, which was distributed along with the questionnaire itself. This was to avoid any breach of confidentiality, as emphasised. The study was conducted with the ethical approval given by the Research and Ethics Committee of UKM (UKM.FPR.SPI 800-2/27). We hereby declare that this study reflects the authors’ own research and analysis in a truthful and complete manner.

## 3. Results

Out of the 455 clinical year undergraduate medical students in UKM, 83.1% (*n* = 378) participated and fulfilled the inclusion criteria. The demographic characteristics of the final survey respondents are summarized in Table 1. Here, 66.1% (*n* = 250) of the respondents were female students, while 33.9% (*n* = 128) were male students. Their ages ranged between 21 and 26 years (mean = 23.13, SD = 1.00). By year of study, 36.8% (*n* = 139) of them were Year 3 students, 29.6% (*n* = 112) were Year 4 students, and 33.6% (*n* = 127) were Year 5 students.

In Figure 1, the prevalence of anxiety and burnout among the clinical year medical undergraduate students in UKM was, respectively, 44.2% (*n* = 167) and 22.2% (*n* = 84).

In Table 2, which compares the prevalence of anxiety by year, Year 3 undergraduate medical students have a higher prevalence of anxiety and burnout, with a significant percentage of 42.5% (*n* = 71) and 41.6% (*n* = 35), respectively, compared with Year 4 and Year 5 undergraduate medical students in UKM. With regards to gender, more female students reported significant anxiety, 67.0% (*n* = 112) and burnout 70.2% (*n* = 59) compared with male students: 33.0% (*n* = 55) and 29.7% (*n* = 25), respectively.

According to Table 3, acceptance coping is the most frequently used (3.13 ± 0.80), while substance abuse was the least favoured (1.12 ± 0.44). These data were measured on a 4-point Likert scale where the frequency of usage of specific coping strategies was rated, with 1 being most infrequent and 4 being most frequent. The mean and standard deviation for the two related items was calculated to represent the scale.

According to Table 4, there was a significant difference in the percentage of students with extremely severe anxiety in the presence and absence of burnout, namely 23.8% (*n* = 20) vs. 4.8% (*n* = 14), with a *p*-value < 0.001.

Because of the nonparametric distribution of the data, the association between the scores of the coping mechanisms and psychological distress was determined using Spearman’s correlation test in Table 5, where an *r* value of <0.20 was considered as a weak correlation, an *r* value of 0.2 to 0.8 was considered as a moderate correlation, and *r* > 0.8 was considered as a strong correlation. Among the three types of coping mechanisms, avoidant coping showed a significantly positive moderate correlation with the presence of anxiety (*p* < 0.001) and the presence of burnout (*p* < 0.001). Meanwhile, students who applied neither approach nor avoidant coping mechanisms also had a significant correlation with the presence of burnout with a *p*-value of 0.001; however, this correlation is a positive weak correlation.

## 4. Discussion

In this study, the prevalence of anxiety among the clinical year medical undergraduate students was 44.2% (*n* = 167), while the prevalence of burnout among the students was 22.2% (*n* = 84), with female students reporting higher levels of anxiety and burnout compared with their male counterparts. Our research finding is supported by many emerging studies in the literature, revealing differences based on gender, where women show higher levels of anxiety in response to the COVID-19 pandemic [31,32]. Undergraduate students at university have also been observed to be more fearful of COVID-19 than graduates [33]. If we compare these results with those from the pre-pandemic era from a study by Tohid et al. in 380 UKM medical undergraduate students, where anxiety levels were reported to be 2.4% (*n* = 9), this is a steep increase and may be attributed to several factors [34].

Since the pandemic started, the Malaysian government has enforced strict measures to combat its spread. In March 2020, these rules included a complete restriction on movement and assembly, and closure of public and private institutions of higher learning. These restrictions put a dent in the normal routines of UKM medical undergraduate students, as they had to continue their education at home via digital learning, away from the clinical setting [35].

According to the Centers for Disease Control and Prevention (CDC), some common factors that might add to anxiety and stress levels during a pandemic include concern about the risk of being exposed to the virus, lacking access to the tools and equipment needed to accommodate digital learning, feelings of guilt about not being on the frontline, uncertainty about the future, learning new communication tools, dealing with technical difficulties, and adapting to a different learning space and/or class schedule [36].

Factors influencing the students’ anxiety could also be related to the COVID-19 stress scale (CSS), which identified five factors of stress and anxiety symptoms related to the COVID-19 pandemic. They include fears of danger and contamination, fears about economic consequences, Coronavirus-related xenophobia, compulsive checking and reassurance seeking, and traumatic stress symptoms [37].

For UKM’s clinical year medical undergraduate students, transitioning from more conventional methods of learning to digital learning also required certain prerequisites. Students who lacked the elements to accept the migration of their medical education to digital learning, and whose medical training through clinical rotations was suspended became increasingly weary of their level of preparedness, leading to increased levels of anxiety.

Some factors affecting the level of preparedness included a nonconducive environment for digital learning such as unstable internet access (e.g., inadequate mobile data plans or poor Wi-Fi connectivity). Some students may lack electronic devices with adequate processing power and hardware that can handle the hours of learning, the demands of running video conferences or running multiple applications and browsers at a time, and so on. On top of that, students were also required to learn to navigate and troubleshoot issues when using certain software as well as hardware (e.g., microphone or webcam issues) to have the most positive experience in class [38]. Experiencing such issues while not receiving technical support for them may result in increased frustration and stress.

Another factor for the increased level of anxiety could be related to regular social media use during the pandemic, where the high daily rates of new cases and deaths, and the information and misinformation overload via social media could lead to the development of anxiety and mood disorders. Moreover, students probably had increased screen time in conjunction with the migration to digital learning. Students tended to spend more time on electronic devices, and hence were more prone to social media exposure. The social panic caused by COVID-19 and the public health emergency measures may have led to anxiety, alongside a series of other negative effects such as insecurity, emotional isolation, stigmatisation, and economic loss [39,40,41,42]. Prolonged screen time is also associated with psychological, cognitive, and musculoskeletal impairments [39,40,41].

Considering the students holistically, there was also the aspect of economic changes in the context of the pandemic. According to the Department of Statistics Malaysia (DOSM), an unemployment rate of 5.3% was reported in May 2020 during the pandemic, compared with 3.3% in 2019 before the pandemic. The impacts of COVID-19 on Malaysia’s economy can also be seen via the depreciation in the Malaysian Ringgit against USD [43]. In UKM, the undergraduate medical program is a full-time course, and hence students are unemployed. Many students depend financially on scholarships, student loans, or their families. Fees and daily living expenses still required payment, and with the economic impact of the pandemic on many families, the students consequently faced increased financial burdens, stress, and conflict [35,38,44].

Psychological distress could also be due to the quarantine and lockdown measures. These measures might cause significant changes to students’ social network behaviour and mental health, e.g., increased social media exposure, internet addiction, poor sleep quality, increased loneliness, and a sense of estrangement from family and peers. Students might also experience increased family conflicts from needing to communicate with their families more often at close quarters when quarantining at home [45].

The overall level of burnout during the pandemic (22.2%, *n* = 84) was less than before the pandemic, according to data from a study in Universiti Sains Malaysia (67.9%, *n* = 307) during the pre-pandemic period as a comparison [46].

In this study, Year 3 medical students reported higher levels of burnout (25.2%, *n* = 35) compared with Year 4 (20.5%, *n* = 23) and Year 5 students (20.5%, *n* = 26). Year 3 students also reported higher levels of anxiety ((51.1%, *n* = 71) compared with Year 4 (42.0%, *n =* 47) and Year 5 students (38.6%, *n* = 49). This could be due to the students developing better coping skills over the years. It could also be due to the Year 3 students undergoing a major adaptation to the intense clinical responsibility. Furthermore, the Year 4 and Year 5 students had gone through their psychiatry posting, which could have given them better awareness and insights into mental health issues, and helped instil healthier attitudes in them [47]. Students may also have developed a better support network over the years with their friends, lecturers, and mentors, having spent more time together over the years. Some students who are emotionally vulnerable may also find their early clinical years more stressful compared with their peers [48,49,50,51].

In this study, there was a significant number of participants with extremely severe anxiety and burnout concurrently (23.8%, *n* = 20). In the late 1990s, researchers began to suggest a link between anxiety and burnout. Some factors contributing to this association could be an individual’s personality traits, where low extroversion and emotional instability could affect an individual’s tendency to burn out. It was also reported that low extroversion is positively correlated with anxiety, while emotional instability has been shown to be positively related to the core component of burnout, i.e., emotional exhaustion, and depersonalization. Individuals who are more extroverted and more emotionally stable are less likely to develop burnout and vice versa [52,53]. Conversely, it was found that factors affecting high burnout levels such as increased job demands, cynicism, and emotional exhaustion were associated with high anxiety levels. The interaction between work situations and individuals’ personalities can create a state of anxiety and, by extension, contributes to the onset of burnout. In short, these studies managed to draw a significant link between anxiety and burnout; however, the exact direction of this relationship (i.e., whether burnout leads to anxiety or vice versa) has yet to be established [53].

Coping mechanisms refer to the specific efforts, both psychological and behavioural, that humans apply to overcome or minimize stressful events [54]. There are four main categories, namely problem-focused, emotion-focused, meaning-focused, and social-focused mechanisms. Problem-focused coping is when one addresses the problem that is causing distress. Emotion-focused coping involves reducing the negative emotions related to the stress. Meaning-focused coping is when one reasons out the problems and understands the meaning of the situation. Social-focused coping is when one seeks emotional support from the community. Coping mechanisms can also be divided into approach coping mechanisms, avoidant coping mechanisms, and neither approach nor avoidant coping mechanisms [6,55]. There is also evidence provided by studies on the role of coping mechanisms in work-related situations. For instance, the coping mechanisms applied when encountering hardships or problems can predict the level of anxiety, depression, and stress among students [56].

It was also reported that there was a difference in coping mechanisms between males and females. It was shown that women use emotion-focused coping more than men. Coping strategies such as high self-blame and less positive reframing in women has shown a positive association with anxiety in women, whereas there was no such effect in men [57].

In this study, it was found that the majority of the students favour using approach coping mechanisms, namely, active coping, emotional support, acceptance, informational support, positive reframing, and planning, rather than avoidant coping mechanisms, namely, behavioural engagement, self-distraction, denial, venting, substance abuse, and self-blame. These results reflect the frequency of students using these coping mechanisms, as self-reported using a 4-point Likert scale [58].

Avoidant coping mechanisms showed a significant positive moderate correlation with the presence of anxiety (*p* < 0.001, *r* = 0.396). Avoidant coping has been previously linked to anxiety, whereas active and problem-focused strategies have been associated with better health outcomes [59].

On the other hand, stressful environments are closely related with poor mental health, physical illness, mediocre performance, and substance abuse. One of the most worrying phenomena is substance abuse among health care practitioners (HCP), as seen in a study from the United Kingdom [56]. Substance abuse was the least favoured coping mechanism in our research, although under-reporting cannot be ruled out. It was lower than in similar studies conducted among medical students in the United Kingdom and North India, where substance abuse is commonly used as a coping mechanism to relieve psychological stress [50,60]. The lower prevalence rate of substance abuse in this study could be explained by the students’ religious beliefs; for example, the consumption of alcohol and illicit drugs, including inhaled substances such as smoking, hookahs, and vaping, are prohibited among Muslim people, who comprised 57.4% (*n* = 217) of the participants [61,62]. Despite the low rate of self-reported substance abuse in this study, individuals are prone to develop serious mental health issues by applying other negative coping mechanisms because they might lead to unresolved issues.

Students who applied avoidant coping mechanism during the COVID-19 pandemic showed a significant positive moderate correlation with the presence of burnout. This result was reflected by a study on UK doctors, where avoidant coping mechanisms (e.g., self-distraction and self-blame) were more frequently used by those suffering from burnout. This suggests that avoidant coping mechanisms significantly contribute to the development of burnout [63,64].

There was also a significant positive weak correlation between neither approach nor avoidant coping mechanisms and the presence of burnout. However, upon a detailed analysis on individuals using neither approach nor avoidant coping mechanisms, it was found that the coping mechanism of humour had a positive weak correlation with the presence of burnout (*p* < 0.001, *r* = 0.198). This was reflected by a study on correctional officers in offender centres in the US, where humour was related to an increased level of emotional exhaustion and depersonalisation, both of which are components related to burnout [65].

In general, problem-focused coping is the best coping mechanism. This is because it removes the root problem of the stressor and finds a long-term solution to the problem. However, not all problems can be coped with by using problem-focused coping. This is when emotion-focused coping plays a role, such as acceptance of the situation [66].

## 5. Conclusions

The prevalence of anxiety and burnout was concerning. Increased anxiety and burnout among students may negatively impact aspects of their personal, professional, and academic lives. The findings of this study can be used by the academic or faculty administrators for the implementation and improvement of preventive measures specifically for anxiety and burnout among medical students in order to eliminate or minimize the unwanted consequences of anxiety and burnout on physical and mental health, social development, competency, and academic performance among medical students. Early recognition, coping skills, and preventive measures should be emphasised to faculty members and students as a joint effort to prevent the negative ramifications.

### 5.1. Limitations

The use of self-reporting questionnaire has limitations, as they are subject to method variance effects and response biases, such as socially desirable responding. The results of this study are also limited to the clinical year medical undergraduate students from UKM; therefore, the results cannot be generalized to a larger population. Furthermore, this study was limited in its monitoring and reporting aspects. Because of the ethical considerations and the current circumstances of this study, intervention was difficult without compromising the anonymity of the participants.

### 5.2. Recommendations

It is important to provide students with a holistic medical education emphasizing positive coping mechanisms and building resilience, on top of competency in the clinical skills [56]. Students with good coping skills have a higher success rate in tackling their problems. They also appear to be more confident, as they have a sense of control over their problems [67]. In a study carried out in the US, it was found that only one-third of medical students with burnout seek help because of perceived stigma and negative personal experiences. Medical students from that study reported that they had observed faculty staffs and peers breaching the confidentiality of other students’ mental health problem and engaging in discriminatory behaviour towards students with emotional problem [68]. Therefore, medical schools could tackle the stigma attached to mental health problems and the barriers to seeking help by educating faculty staff about the confidentiality policies and procedures, and by monitoring and responding to reports of discrimination due to mental illness [69]. An increased sense of personal accomplishment was found after the implementation of a mentorship program. Both mentors and mentees viewed the program positively and perceived multiple benefits [51].

## Figures and Tables

**Figure 1 ijerph-19-13010-f001:**
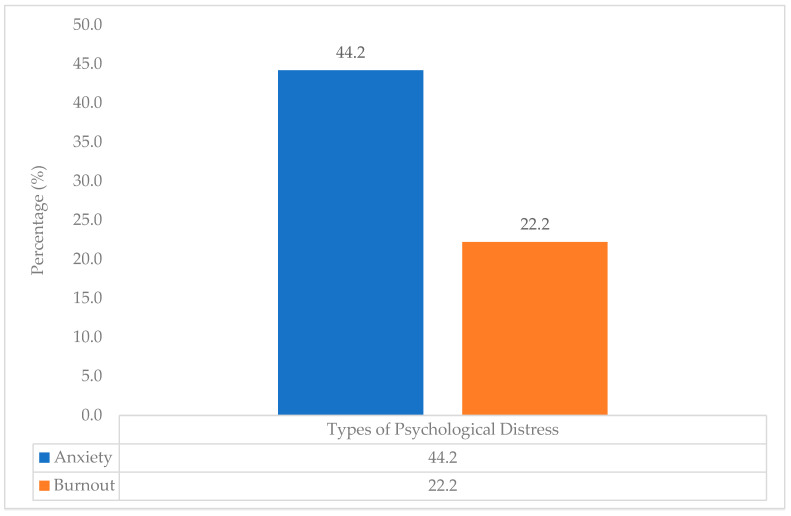
Prevalence of anxiety and burnout among clinical year medical undergraduate students in UKM during the COVID-19 pandemic.

**Table 1 ijerph-19-13010-t001:** Sociodemographic characteristics of the respondents.

Characteristics	*n* = 378 (%)
**Gender**	
Male	128 (33.9)
Female	250 (66.1)
**Ethnicity**	
Chinese	66 (17.5)
Indian	76 (20.1)
Malay	217 (57.4)
Others	19 (5.0)
**Year of study**	
Year 3	139 (36.8)
Year 4	112 (29.6)
Year 5	127 (33.6)
**Age (mean: 23.13 ± SD 1.00)**	
21	6 (1.6)
22	110 (29.1)
23	122 (32.3)
24	113 (29.9)
25	32 (5.6)
26	6 (1.6)

**Table 2 ijerph-19-13010-t002:** Association of psychological distress with sociodemographic characteristics among the respondents.

	Anxiety (*n* = 167)			Burnout (*n* = 84)		
	*n*	%	*p*-Value	*n*	%	*p*-Value
**Gender**			0.734			0.368
Male	55	33.0		25	29.7	
Female	112	67.0		59	70.2	
**Year of study**			0.104			0.573
Year 3	71	42.5		35	41.6	
Year 4	47	28.1		23	27.3	
Year 5	49	29.3		26	31.0	

**Table 3 ijerph-19-13010-t003:** Measures of the coping dimensions implemented.

Coping Dimensions	Mean ± SD
Acceptance	3.13 ± 0.80
Religion	3.09 ± 0.94
Self-distraction	3.06 ± 0.81
Positive reframing	3.02 ± 0.84
Active coping	3.01 ± 0.78
Planning	2.89 ± 0.83
Emotional support	2.71 ± 0.90
Informational support	2.61 ± 0.88
Venting	2.39 ± 0.82
Humour	2.27 ± 0.98
Self-blame	2.25 ± 0.90
Behavioural disengagement	1.79 ± 0.78
Denial	1.58 ± 0.69
Substance abuse	1.12 ± 0.44

**Table 4 ijerph-19-13010-t004:** The association between anxiety and burnout in clinical year medical undergraduate students in UKM during the COVID-19 pandemic. * *p* < 0.05 is considered as significant.

	Severity of Anxiety, *n* (%)					
	Normal	Mild	Moderate	Severe	Extremely Severe	*p*-Value
Presence of burnout						0.001 *
Yes	26 (31.0%)	10 (11.9%)	14 (16.7%)	14 (16.7%)	20 (23.8%)	
No	185 (62.9%)	51 (17.3%)	28 (9.5%)	16 (5.4%)	14 (4.8%)	

**Table 5 ijerph-19-13010-t005:** Spearman’s correlation test for the associations of anxiety and burnout with the coping mechanisms implemented.

Coping Mechanism	Anxiety		Burnout	
	*r*	*p*-Value	*r*	*p*-Value
*Approach*	0.029	0.579	−0.012	0.821
*Avoidant*	0.396	<0.001	0.341	<0.001
*Neither approach nor avoidant*	0.086	0.097	0.176	0.001

## Data Availability

The data supporting the findings of this study are available from the corresponding author upon reasonable request, subject to approval by the local institution.

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
