# Peer review of "Prevalence of Anxiety and Burnout, and Coping Mechanisms among Clinical Year Medical Undergraduate Students in Universiti Kebangsaan Malaysia Amidst the COVID-19 Pandemic"

_ijerph, 2022, doi:10.3390/ijerph192013010_

Round 1

Reviewer 1 Report

Dear authors,

 It was such a pleasure reviewing your well presented work. The topic addresses by this manuscript deserves be explored, it is very relevant to the health of future medical professionals.

The abstract and introduction sections are well written and provide an appropriate support to the topic explored by the current manuscript.
Also the Methodology section
 is clear and presents an adequate study design.

Results section also remains appropriate showing the outcomes from the analysis carried out based on the proposed methodology. The formatting of tables and figures should be standardized. Also in Figure 1 (page 5) the title (information already contained in the legend) and the graph gridlines should be eliminated.

Conclusions are clear and adequate to the aim established for this study.

Author Response

I would like to express my sincere gratitude for your valuable and constructive review of our paper. The comments have helped to further strengthen the overall quality of the manuscript. With utmost care, I have incorporated all the suggestions/corrections as proposed. The specific responses to comments are listed in the attachment.

Reviewer 2 Report

Line 28 - consider changing 'will' to may

Line 58 - spacing in 'anew'

Line 74  - should registered tradenames be identified (Zoom and Team)?

Line 121-123 - Brief intro of DASS-21, CBI and Brief-COPE is needed here

Table-1 and Caption: Align to the center of the page

Figure-1 and Caption: Align to the center of the page

Table-2 and Caption: Align to the center of the page

Table-3 and Caption: Align to the center of the page

Table-3: What is the scale/range for this dataset?

Table-4 and Caption: Align to the center of the page

Table-5 and Caption: Align to the center of the page

Line-210: is n=9 sample comparable?

Line 242 - white space

Line 250 "The social panic caused by COVID-19" in this context of media exposure induced natural disaster related disorders, the authors can also refer to Dr. Betty Pfefferbaum's work.

Line 261 - suggest rephrase to 'students financially depend'

Line 266 - what is 'that' here?

Line 276-278: consider separating out burnout and anxiety data, unless a correlation is implied (i.e. students with both) as shown in Line 290

Line 337 - reference for Likert scale?

Line 350-352 - does religion also rule out inhaled substances such as smoking, hookah, and vaping?

Line 365 - only humor coping not described 

Line 377 - 'will' or 'may'?

Line 378 - cannot be definitively stated in absence of causation

Line 383 - extra white space

Line 390-392: awkward, consider rephrasing

Author Response

(The authors gave the same response as above.)

Reviewer 3 Report

The paper is interesting but there are many drawbacks:

- It lacks a clear theoretical framework;

- The time interval is too short to measure long-term effects;

- There are no treatment / control groups, no pre-measurement among the population. 

Despite its qualities, I do not see novelty in the paper. I would suggest the authors to turn it into an experiment or to relate the levels of rigidity (see Response2covid19 or the OxGRCT) to the behavior of students.

Author Response

(The authors gave the same response as above.)

Round 2

Reviewer 3 Report

You did not answer to the following comments :

- It lacks a clear theoretical framework;

- There are no treatment / control groups, e.g. do students behave diffeeently ?

- I suggested the authors to turn it into an experiment or to relate the levels of rigidity (see Response2covid19 or the OxGRCT) to the behavior, but they did not seem to understand my comment , I.e. do you see any changes across time?
